# False Myths versus Medical Facts: Ten Common Misconceptions Related to Dry Eye Disease

**DOI:** 10.3390/biomedicines8060172

**Published:** 2020-06-24

**Authors:** Giuseppe Giannaccare, Vincenzo Scorcia

**Affiliations:** Department of Ophthalmology, University Magna Græcia of Catanzaro, Viale Europa, Germaneto, 88100 Catanzaro, Italy; vscorcia@unicz.it

**Keywords:** dry eye, dry eye disease, misconceptions, medical facts

## Abstract

Since the first definition of dry eye, rapid progress has been made in this field over the past decades that has guided profound changes in the definition, classification, diagnosis and management of the disease. Although dry eye is one of the most frequently encountered ocular conditions, various “old” misconceptions persist, in particular among comprehensive ophthalmologists not specialized in ocular surface diseases. These misconceptions hamper the correct diagnosis and the proper management of dry eye in the routine clinical practice. In the present review, we described the 10 most common misconceptions related to dry eye and provided an evidence-based guide for reconsidering them using the format “false myth versus medical fact”. These misconceptions concern the dry eye definition and classification (#1, #2, #3), disease physiopathology (#4), diagnosis (#5), symptoms (#6, #7) and treatment (#8, #9, #10). Nowadays, dry eye is still an under-recognized and evolving disease that poses significant clinical challenges to ophthalmologists. The two major reasons behind these challenges include the heterogeneity of the conditions that fall under the umbrella term of dry eye and the common discrepancy between signs and symptoms.

## 1. Introduction

Dry eye is one of the most frequently encountered ocular conditions affecting a large part of the population, ranging from 20% to 50% [1]. As the population ages, this number is likely to increase. Given the high prevalence, dry eye represents a major public health issue that has effects on patient quality of life, health care resources and the economy. In addition, the symptoms and signs of dry eye are some of the most common causes of patients’ dissatisfaction in the ophthalmic practice.

Since the first comprehensive definition of dry eye was published in 1995 on the basis of consensus from the National Eye of Institute (NEI) Industry Working Group [2], rapid progress in basic and clinical research has been made in this field over the past three decades that reflects profound changes in the definition, diagnostic criteria and management strategies of the disease [3].

Despite these continuous advances, various “old” misconceptions seem to persist when approaching dry eye care, particularly among comprehensive ophthalmologists that might be not inclined and/or equipped to overcome them. Reconsidering these misconceptions is an important challenge because, due to the high frequency and the chronic/recurrent course of the disease, patients with dry eye (in particular those affected by milder forms) often are managed by comprehensive ophthalmologists rather than by ocular surface specialists.

This review aims at providing an evidence-based guide for reconsidering 10 major misconceptions related to dry eye in the format “false myth versus medical fact”.

## 2. From False Myths to Medical Facts

### 2.1. False Myth #1: “Dry Eye Is Just a Disorder or a Dysfunction”

In the first definition from the NEI Working Group, dry eye was described as a disorder of the tear film resulting from a simple imbalance between tear production and evaporation [2]. The understanding of this condition improved significantly over time, and a consensus on a new definition was achieved in 2007 at the first International Dry Eye Workshop (DEWS) sponsored by the Tear Film and Ocular Surface Society (TFOS) that introduced for the first time the term “disease”, referring to dry eye [4]. In the recently updated definition, the term disease was retained for the important implications on the perspective of this condition from the sides of the patient, doctor and healthcare system [5]. It should be pointed out that dry eye disease (DED) can occur both as an isolated condition of the ocular surface or as part of a systemic disorder, requiring the assistance of internists in a multidisciplinary patient-centered approach [6]. Recently, the awareness of the DED association with hormonal changes and systemic drugs is increasing among physicians in other specialties [7]. In particular, taking multiple medications (polypharmacy) has been shown to represent a risk factor for DED.


**Medical Fact #1: “Dry Eye Is a Multifactorial Disease Occurring both as Isolated or as Part of a Systemic Condition”**


### 2.2. False Myth #2: “Dry Eye Is a Disease of the Cornea, Lids, Tears, etc…”

The first mention of ocular surface was made in 1977, when Thoft and Friend described the morphofunctional connection between corneal and conjunctival epithelia and their interaction with the tear film [8]. By the mid-2000s, the understanding of this complex set of structures reached a crucial point with the identification of the “Lacrimal Function Unit” and the “Ocular Surface System” [4,9]. The latter includes cornea, conjunctiva, main and accessory lacrimal glands, meibomian glands, eyelashes with their associated glands of Moll and Zeis and the nasolacrimal duct. All these components of the system are linked functionally by the continuity of the epithelia, by innervation and by the endocrine, vascular and immune systems [9]. Of these, the neural feedback loop regulates ocular surface lubrication, with ocular sensation through corneal innervation driving basal and reflex tear production by the lacrimal gland. There is now evidence that any abnormality of the ocular surface can trigger a disequilibrium in all the other components of the system. Indeed, treatments for ocular surface disease should consider the ocular surface system as a whole, rather than target any single component of the system alone. This will be able to promote the restoration of ocular surface homeostasis.


**Medical Fact #2: “Dry Eye Affects the Whole Ocular Surface System”**


### 2.3. False Myth #3: “Dry Eye Is Caused by a Lack of Tears”

In the past, dry eye was thought to be merely caused by aqueous tear insufficiency. The first definition of dry eye (NEI Report) as well as the subsequent revision (TFOS DEWS) distinguished two different separated subtypes, namely aqueous-deficient and evaporative [2,4]. The former is encountered in patients with Sjögren syndrome and graft-versus-host disease [10,11]. The latter is most often attributable to insufficient secretion of the lipid component of the tear film. It is now understood that evaporative and mixed (evaporative plus aqueous-deficient in the same patient) forms account for the majority of overall DED cases [5]. Furthermore, in contrast to what is suggested by the terminology itself, evaporative DED can sometimes result in ocular surface irritation with a secondary increase in tear production and even epiphora (“wet” dry eye) [12]. Recently, the Asia Dry Eye Society proposed a new classification of dry eye based on the concepts of tear film-oriented diagnosis (TFOD), and suggested that there is a third type of dry eye characterized by decreased wettability due to a deficiency in membrane-associated mucin [13].


**Medical Fact #3: “Dry Eye Is Caused by a Lack of Adequate Tears”**


### 2.4. False Myth #4: “Corneal Pain without Stain: Is It Real? No”

“Corneal pain without stain: is it real?” was the title of an intriguing article published in 2009 in which Rosenthal and co-authors integrated the knowledge and principles from other branches of science (neurology, neurobiology, pain physiology) for increasing awareness and encouraging interest in the under-recognized field of corneal pain in the absence of vital dye staining [14]. In fact, in the ophthalmic routine practice, it is not uncommon to be faced with a challenging dilemma: some patients may refer severe ocular discomfort symptoms with no/mild signs detected on slit lamp examination. This discrepancy can be explained, at least in part, by the lack of consistent results of the clinical tests used, the subjective nature of symptoms and individual variations in pain thresholds. This scenario is particularly diagnosed after ocular surgery with injury to the peripheral sensory nerves of the cornea [15]. The subsequent studies on corneal nerves, thanks also to the widespread use of in vivo confocal microscopy, allowed to identify a new condition that falls under the umbrella term of dry eye and is called neuropathic pain [16]. This is a condition characterized by ocular pain, often accompanied by light sensitivity, dysesthesia and photoallodynia. Pain generation can initiate at different steps of the nerve connection between the eyes and brain: nerves on the ocular surface (cornea and conjunctiva), secondary and tertiary neurons, or both. When central neurons are involved, the origin of pain is not from the eye itself but from higher order neurons that connect the eye to the brain [17]. The presence of unaddressed nerve dysfunction should be suspected in all individuals with painful dry eye whose symptoms do not improve after conventional therapies. The use of new compounds with activity on transient receptor potential channels should be considered in these patients [18]. For this task, also biological blood-derived tear substitutes obtained from either patients themselves or donors can be of further help [19,20].

Finally, the awareness and understanding of the role of corneal nerves in DED have increased substantially in the last DEWS II Report that acknowledged “neurosensory abnormalities” as an etiologic factor of dry eye, and listed them in the main definition [5]. The DEWS II classification scheme considers among DED patients not only cases with both signs and symptoms of dry eye but also cases where ocular discomfort symptoms are present without evidence of obvious signs (neuropathic pain) or where marked signs are present in the absence of dry eye symptoms (neurotrophic condition).


**Medical Fact #4: “Corneal Pain without Stain: Is It Real? Yes”**


### 2.5. False Myth #5: “The Diagnosis of Dry Eye Is Time-Consuming and Requires High-Tech Equipment”

Although the detection of the presence of DED is fast and simple, the assessment of the main pathogenic factor(s) contributing to the vicious circle of dry eye in the single clinical picture is often considered time-consuming and challenging, especially for general ophthalmologists who often do not have access to emerging sophisticated technologies. A three-step procedure can be a simple method for reaching the diagnosis of dry eye in 10 min in a low-tech office: (i) collect the patient’s history by means of validated questionnaires and triage questions; (ii) observe without aid devices (e.g., skin, blinking) and with slit lamp biomicroscopy for the evaluation of inflammatory and/or eyelid disease; and (iii) perform first-level tests (break-up time, Schirmer test, vital dye staining, aesthesiometry) (Figure 1). A correct diagnosis able to determine the major causative factor(s) behind DED and assess disease severity is crucial for two main reasons, among others: (i) it will represent the basis for the choice of an appropriate therapeutic strategy [21]; and (ii) individuals who receive a diagnosis of DED had a significantly better DED-related quality of life (QoL) compared with those who did not [22]. In fact, a recent paper demonstrated that DED patients belonging to the undiagnosed group had a significantly worse score of DED-related QoL compared with patients who received a definitive diagnosis. The authors speculated that this could be explained by the fact that patients with undiagnosed DED may perform self-care measures (e.g., use of over-the-counter eye drops, increase in blinking, use protection glasses, warm compress/eyelid hygiene, sleeping longer, exercise, use of oral supplements) inappropriately.


**Medical Fact #5: “The Diagnosis of Dry Eye (Detection & Assessment) Is Possible in 10 min in a Low-Tech Office”**


### 2.6. False Myth #6: “Dry Eye Patients without Corneal Damage Do Not Suffer from Visual Disturbance”

Traditionally, dry eye without any corneal damage was considered a condition causing just discomfort symptoms, but without detrimental effects on visual function [23]. The last decades of research have revealed that dry eye with an unstable tear film affects the quality of vision as well, altering light deflection/scattering with the formation of a blurred image on the retina [24,25,26,27]. In fact, since the tear film is the first entry point of light, it has to be smooth during the entire interblink interval to guarantee an optimal visual acuity. As a consequence, the 2007 TFOS DEWS Report listed “visual disturbance” as one of the main symptoms of DED and included its mention in the main definition [4]. Therefore, there has been a growing emphasis in DED patients to measure not only the quantity of visual acuity using conventional charts, but also the quality of vision by means of contrast sensitivity, light scattering and aberrations [28]. Furthermore, the impact of DED on visual function is evaluated also thanks to the validated questionnaire Ocular Surface Disease Index (OSDI), which includes six questions related to visual disturbance (blurred or poor vision) and visual function (problems during reading, driving, working on a computer, watching TV) [29].


**Medical Fact #6: “Dry Eye Patients Suffer from Visual Disturbance Regardless of Corneal Status”**


### 2.7. False Myth #7: “Dry Eye Disease does not Influence Surgical Outcomes”

Despite that modern ocular surgery (mainly refractive and cataract) has reached high standards of safety and efficacy, patients’ satisfaction is not always consistent with these advances. The main reason is related to ocular discomfort symptoms that patients frequently report in the operated eye [30]. These symptoms usually appear a few weeks after surgery and then gradually decrease over the subsequent months as a result of the natural healing process [31]. However, a certain population of post-surgical patients demonstrated persistent dry eye signs and/or symptoms, even several years after the operation [32]. Therefore, there is a growing need to encourage the preoperative assessment of the risk of DED development or worsening in all patients undergoing surgery in order to improve both patient and surgeon satisfaction. A complete assessment of the ocular surface before surgery is essential to stratify patients at risk for DED, to warn them about the possible onset of postoperative symptoms and to institute the appropriate treatment as soon as possible. Furthermore, undiagnosed or untreated ocular surface disease before surgery is likely to lead to suboptimal postoperative vision quality and quantity due to the inaccuracy of preoperative calculations [33].

Dry eye is highly prevalent also in patients with glaucoma under medical therapy, mostly due to the presence of preservatives that are toxic to the ocular surface. The severity of ocular discomfort symptoms increases with the number of medications used, and has a negative impact on treatment compliance and quality of life [34]. Furthermore, it has been also demonstrated that the presence of ocular surface disease worsened not only surgical outcomes but also intraocular pressure control [35]. Therefore, an adequate management of ocular surface disease is crucial in glaucoma patients in order to improve long-term outcomes of the disease. The first-line therapy includes the use of preservative-free hypotensive medications and tear substitutes; hot compresses to the eyelids and lid massage could be of further help.


**Medical Fact #7: “Optimization of the Ocular Surface Is Crucial for Improving Overall Surgical Outcomes”**


### 2.8. False Myth #8: “All Tear Substitutes Are the Same”

Tear substitutes continue to represent the mainstay of DED treatment, at all stages of severity. They comprise a wide variety of products that differ for viscosity and composition, and typically aim at targeting one or more layers of the tear film [36]. In an ongoing effort to ameliorate tolerability and increase the duration of action, tear substitutes have undergone numerous improvements since the first generation that was simply a saline-based isotonic or hypotonic solution with preservatives. Subsequent generations added natural and synthetic polymers with gentler preservatives or were even preservative-free. Newer generations include osmoprotectants that counteract tear hyperosmolarity, lipid oil-in-water nano-emulsions that improve the residence time on the tear film and multiple-action tear substitutes that combine the action of different polymers for widening the spectrum of action [37]. Although a recent meta-analysis failed to indicate one or more tear substitutes with higher efficacy compared with the other ones [38], the type of the proper tear substitute should be chosen on the basis of the single clinical picture, taking into account the main causative factor(s) underlying the disease (Figure 2). In the case of tear film instability or meibomian gland dysfunction without inflammation, the choice should go towards a complete tear substitute (fluid during the day and gel at night); in the case of meibomian gland dysfunction with inflammation, acute blepharitis or inflammation/tear dysfunction, the choice should go towards a tear substitute with an osmo/bioprotectant; in the case of epithelial damage, the choice should go towards a gel tear substitute; and in the case of irritation, the choice should go towards a multicomponent tear substitute.


**Medical Fact #8: “The Type of Tear Substitute Should Be Chosen According to the Single Clinical Picture”**


### 2.9. False Myth #9: “Tear Substitutes Can Be Used on As-Need Basis”

Individual variability in tear substitutes usage is the norm rather than the exception in the DED routine clinical practice. A number of explanations are possible, including poor doctor–patient communication, different conditions covered by the same diagnosis of DED and the waxing and waning nature of the disease [39]. As shown for other chronic diseases, dry eye requires long-term treatment with maintenance doses of drugs for reaching disease stabilization over time [40]. An observer-masked clinical trial comparing fixed (four times daily) versus as-needed use of tear substitutes confirmed that the former regimen provided better symptomatic relief [41]. Furthermore, when prescribing tear substitutes, patient education regarding purpose of treatment, specific agent chosen and fixed regimen dosage is fundamental to improve compliance rates [42]. This issue appears to be even more important considering that tear substitutes are an over-the-counter preparation and a plethora of different types are available on the market. A recent study showed that more than two-thirds of the patients with persistent DED reported having discontinued their treatment, and that only one tenth of them was using the same type of tear substitute prescribed by the physician during the visit [43].


**Medical Fact #9: “Tear Substitutes Should Be Prescribed and Used with Regular Dosing”**


### 2.10. False Myth #10: “All Corticosteroids Are Equally Risky for Medium/Long Term Use in Severe Dry Eye”

An old conception of DED therapy was that the addition of a tear volume to the dry ocular surface alone should be able to solve the disease. Hence, artificial tears remained the only mainstay of therapy for decades. As inflammation becomes better recognized, before in dry eye owing to Sjögren syndrome and later in primary dry eye without any predisposing systemic conditions, the importance of quantifying and managing it increased over time [44]. According to this paradigm shift, DED ceased to be just a lack of water and became a complex disease involving inflammation. It is now clear that DED can be chronically self-maintained through a cycle of local and systemic responses, which include ocular surface inflammation involving both innate and adaptive immune responses [45]. Inflammation is also strongly linked to tear osmolarity that represents another core mechanism of DED. Therefore, the increasing focus on developing anti-inflammatories to address DED symptoms and signs followed our understanding of the disease pathophysiology. An anti-inflammatory therapy should be used in conjunction with first-line therapies, particularly in moderate to severe forms to break the vicious cycle [36]. Our current toolbox of anti-inflammatory strategies for DED includes cyclosporine A, nonsteroideal anti-inflammatory drugs, corticosteroids and lifitegrast, each product with a different mechanism, potency and latency of action [46,47,48,49,50]. Among these, corticosteroids have a strong anti-inflammatory activity by suppressing the production of pro-inflammatory cytokines as well as by inhibiting a broad range of specific immune responses mediated by T cells and B cells [51]. However, in addition to their therapeutic effect, corticosteroids can produce a plethora of adverse side effects, mainly cataract and intraocular pressure increase, which are related to the drug concentration, chemical formulation and composition of the vehicle [52]. Corticosteroids eye drops can be used according to a pulse (e.g., four times daily for two–four weeks) or a tapered (e.g., three times daily for one week, twice daily for two weeks, once daily for four weeks) scheme. A recent review reported a good quality of evidence for two types of “soft” corticosteroids, namely loteprednol and fluoremetholone, in terms of efficacy in the setting of DED without negative effects on intraocular pressure, even when used in the medium/long term [46]. A double-masked, randomized controlled study showed that also clobetasone butyrate at a low dosage is safe and effective in treating patients with DED owing to Sjögren syndrome [53].


**Medical Fact #10: “Medium/Long Course of Soft Steroids Can Be Prescribed with Safety in Severe Dry Eye”**


## 3. Conclusions and Future Directions

Dry eye is a common, under-recognized and evolving disease that poses significant clinical challenges to ophthalmologists, which is not yet completely answered. It is a disabling disorder affecting both visual function and quality of life with significant socio-economic impact. Given the high prevalence of the disease and the chronic/fluctuating nature of the disease, almost the totality of ophthalmologists faces with the challenges of diagnosing and managing DED cases. Additionally, the aging of the population along with environmental and lifestyle-related factors may further increase the risk of developing DED. Reconsidering the most common misconception related to this disease in light of recent evidence can help physicians in reaching a correct diagnosis of DED that is instrumental for the choice of the proper treatment. However, a recent paper analyzed the evidence gap pertaining important research questions about treatment effectiveness in DED, and found that only some of them have been addressed by reliable systematic review evidence that did not come to definitive conclusions [54].

It is important that new treatments address unmet needs not only from the medical perspective but also from the patient’s one. Since several reports reveal discrepancies between DED signs detected by ophthalmologists and symptoms reported by patients, one of the goals of the therapy should be to alleviate patients’ discomfort, regardless of changes in objective signs. Therefore, future research should be directed toward developing diagnostic measures able to detect biomarkers with higher predictive values of positive response after treatment [55].

In terms of treatment, a new strategy, namely tear film-oriented therapy (TFOT), which made individualized treatment tailored to the patients’ DED conditions according to tear film abnormalities revealed by TFOD, is currently applied in Asia [56]. The TFOT concept has made great progress with the advent of the aqueous/mucin secretagogue which is currently not available worldwide. New drugs with novel mechanisms of action and advances in drug delivery technology are currently under investigation and could be useful for patients with DED who do not respond adequately to available treatments [57]. Among these, DED patients with neuropathic pain are particularly difficult to be successfully treated, and low rates of satisfactory responses have been described with current treatments [58]. Therefore, targeted strategies to approach directly corneal and central nerves are desirable. Finally, there is an increasing body of evidence that life style habits influence the onset and/or the worsening of DED [59], and further research should investigate the role of life style interventions in preventing and/or treating DED.

## Figures and Tables

**Figure 1 biomedicines-08-00172-f001:**
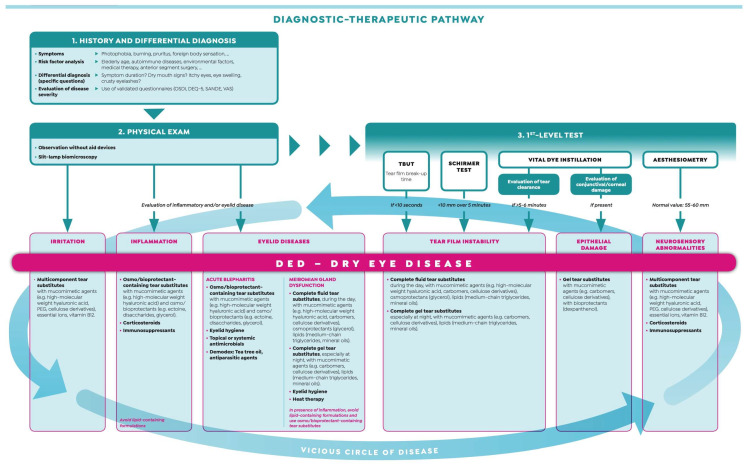
Diagnostic–therapeutic pathway of dry eye disease. Modified from “Malattia dell’Occhio Secco—Dalla definizione alle strategie terapeutiche” with the permission of Editamed srl.

**Figure 2 biomedicines-08-00172-f002:**
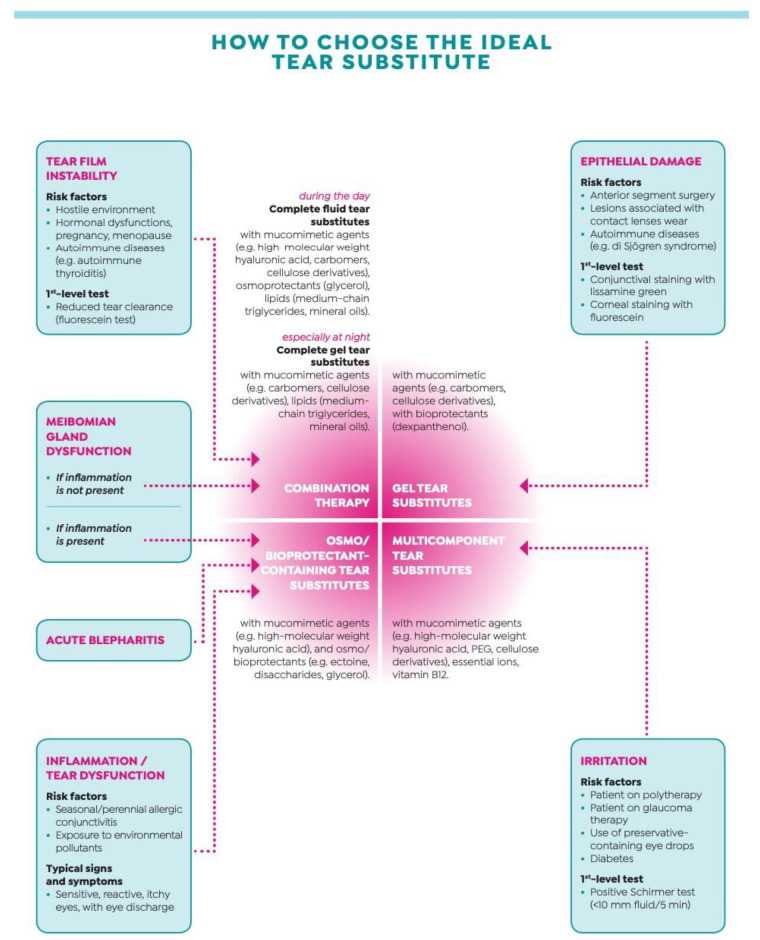
The choice of the ideal tear substitute according to patients’ clinical picture. Modified from “Malattia dell’Occhio Secco—Dalla definizione alle strategie terapeutiche” with the permission of Editamed srl.

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
