# Peer review of "False Myths versus Medical Facts: Ten Common Misconceptions Related to Dry Eye Disease"

_biomedicines, 2020, doi:10.3390/biomedicines8060172_

Round 1

Reviewer 1 Report

I have read with pleasure this direct and concise article by Giannaccare and Scorcia entitled "False Myths versus Medical Facts: Ten Common Misconceptions Related to Dry Eye Disease". I found it well written and even if they did not go into the molecular aspects, the clinic is well structured. However, the authors must describe an important aspect of DED : that concomitant with the use of chronic topical therapies such as in glaucoma. This is because secondary inflammation worsens glaucoma and can lead to surgery for the patient. So they need to describe this and possibly give therapeutic indications. Also among the cortisone drugs to use in DED I find it useful to mention Clobetasone which has few side effects. Finally, a mention should be made of NSAIDs which are also anti-inflammatory drugs that can be used. If correct and slightly enlarged following my indications, the article can be accepted for publication

Author Response

Thanks for the positive comments. We agree with you that an interest aspect of dry eye disease (DED) is related to patients affected by glaucoma under medical therapy. We added the following new paragraph in the false myth n. 7 (page 9 line 9):

“Dry eye is highly prevalent also in patients with glaucoma under medical therapy, mostly due to the presence of preservatives that are toxic to the ocular surface. The severity of ocular discomfort symptoms increases with the number of medications used, and has a negative impact on treatment compliance and quality of life [New ref 34: Zhang et al. Eye Contact Lens. 2019]. Furthermore, it has been also demonstrated that the presence of ocular surface disease worsened not only surgical outcomes but also intraocular pressure control [New ref 35: Batra et al. J Glaucoma. 2014]. Therefore, an adequate management of ocular surface disease is crucial in glaucoma patients in order to improve long-term outcomes of the disease. The first-line therapy includes the use of preservative-free hypontensive medications and tear substitutes; hot compresses to the eyelids and lid massage could be of further help.”

We agree with you that also clobetasone is safe and effective in dry eye. We added the following sentence in the false myth n. 10 (page 13 line 12):

A double-masked, randomized controlled study showed that also clobetasone butyrate at low dosage is safe and effective in treating patients with DED owing to Sjögren syndrome [new ref 53: Aragona et al. Eur J Ophthalmol. 2013].

We agree with you that nonsteroidal anti-inflammatory drugs (NSAIDs) can be useful in resolving ocular discomfort symptoms in DED patients. However, they should be used with caution and under close monitoring for the risk of corneal melting. We listed NSAIDs among the armamentarium of DED therapy (new ref 50: Aragona et al. Eye. 2005).

Reviewer 2 Report

Dry eye disease is a very common but under-recognized disease that poses significant clinical challenges to ophthalmologist. Because it is an evolving and not yet completely answered disease, previous misconceptions may hamper the correct diagnosis and the proper management of this disease in the routine clinical practice. The author summarized 10 false myths versus medical facts based on the current medical evidence. It is considerably helpful for comprehensive ophthalmologists to diagnose and manage dry eye patients. There are only some comments for the author:

  1. Lines 129-132, page 3: "A correct diagnosis able to determine the major causative factor(s) behind DED and assess disease severity is crucial for two main reasons, among others: i) it will represent the basis for the choice of an appropriate therapeutic strategy [21]; ii) individuals who receive a diagnosis of DED had a significantly better dry eye-related quality of life compared to those who did not [22]." The latter statement "ii)" is not very clear, please add some explanation for this sentence.
  2. Lines 142-144, page 4: "In fact, since tear film is the first entry point of light, it has to be smooth during the entire interlink interval to guarantee an optimal visual acuity." Did the author mean "inter-blink"?
  3. Line 152, page 4: "Medical Fact #6: "Dry eye patients suffer from visual disturbance regardless corneal status.”" Please confirm "...regardless corneal status" or "...regardless of corneal status".
  4. Line 153, page 4: "False Myth #7: "The only surgical outcome is postoperative visual acuity.”" This statement is not clear. Many surgeries does not put visual acuity as the primary outcome measurement. Please modify this description.
  5. Lines 199-201, page 7: "An observer-masked clinical trial comparing fixed (four times daily) vs as-needed use of tear substitutes
    confirmed that the former regimen provided better symptomatic relief [39]." This statement seems corresponding to ref. 40 instead of ref. 39. Please carefully recheck your citations.
  6. Lines 211-212, page 7: About this statement "In the past, the traditional understanding was that since dry eye are not wet adding tear volume should solve the problem.", did the author miss some words?
  7. Lines 218-219, page 7: "Inflammation is also strongly linked to tear osmolarity that represents another chore mechanism of DED". Did the author mean "core"?

Author Response

Thanks for your positive comments. Please find below the replies for each point:

  1. We agree with you that a further sentence is useful to better explain this point, as follows: “In fact, a recent paper demonstrated that DED patients belonging to the undiagnosed group had a significantly worse score of DED-related QoL compared to patients who received a definitive diagnosis. The authors speculated that this could be explained by the fact that patients with undiagnosed DED may perform self-care measures (e.g. use of over-the-counter eye drops, increase in blinking, use protection glasses, warm compress/eyelid hygiene, sleeping longer, exercise, use of oral supplements) inappropriately.”
  2. Yes, sorry for this language error. We corrected the text accordingly.
  3. Yes, sorry for this language error. We corrected the text accordingly.
  4. We agree with you and changed the false myth n. 7 as follows (page 9 line 1): “Dry eye disease does not influence surgical outcomes.”
  5. Sorry for the mistake. We corrected the reference list accordingly. Furthermore, we checked also all the references of the paper and we confirm that the new list is correct.
  6. We rephrased the sentence in question in order to make clearer this issue (page 12 line 12): “An old conception of DED therapy was that the addition of tear volume to the dry ocular surface should be able as alone to solve the disease.”
  7. Sorry for the typo. We corrected the text accordingly.